# Taming active turbulence with patterned soft interfaces

P. Guillamat[1,2], J. Ignés-Mullol[1,2] & F. Sagués[1,2]

Active matter embraces systems that self-organize at different length and time scales, often exhibiting turbulent flows apparently deprived of spatiotemporal coherence. Here, we use a layer of a tubulin-based active gel to demonstrate that the geometry of active flows is determined by a single length scale, which we reveal in the exponential distribution of vortex sizes of active turbulence. Our experiments demonstrate that the same length scale reemerges as a cutoff for a scale-free power law distribution of swirling laminar flows when the material evolves in contact with a lattice of circular domains. The observed prevalence of this active length scale can be understood by considering the role of the topological defects that form during the spontaneous folding of microtubule bundles. These results demonstrate an unexpected strategy for active systems to adapt to external stimuli, and provide with a handle to probe the existence of intrinsic length and time scales.

[1] Department of Materials Science and Physical Chemistry, Universitat de Barcelona, Barcelona, 08028 Catalonia, Spain. [2] Institute of Nanoscience and Nanotechnology (IN2UB), Universitat de Barcelona, Barcelona, 08028 Catalonia, Spain. Correspondence and requests for materials should be addressed to F.S. (email: jordi.ignes@gmail.com)

Many examples of either living or activated entities display intriguing modes of non-equilibrium behavior that cover a wide range of length scales and obey vastly different operational mechanisms. This general framework defines what is referred to as the realm of active matter[1, 2]. Flows in active systems are intrinsically autonomous and their lack of spatial and temporal coherence seem to preclude any sort of external controlling intervention. This vision is supported by the emergence of the flow regime known as active turbulence[3–8], which determines the collective behavior in bacterial colonies[9–11] or cytoskeletal networks[12–14], and has a major role in chemical transport and nutrient distribution[9].

In spite of recent theoretical[15–17] and experimental[18–20] attempts, less attention has been dedicated to analyzing the role of confining conditions on active fluids. This is an important issue in a cell biology context, for instance in relation to cytoplasmic streaming in eukaryotes[21]. In fact, recent experiments with active biofluids have revealed that strong coupling with substrates can lead to the steering of otherwise chaotic flows[11, 22]. Boundary conditions can determine the configuration of soft materials, such as colloidal suspensions and liquid crystals. Yet, a fundamental question is how active materials, endowed with nonequilibrium length and time scales, adapt to confinement, and whether external constraints can be used to modify the behavior of active systems in a controlled manner, for instance in a microfluidic lab-on-a-chip context.

In vitro reconstitutions of the cytoskeletal network[12] provide an excellent model active material to address such fundamental issues without the intricacies of a living system. A recently developed tubulin-kinesin gel has been shown to organize as an ordered active material when condensed at a soft interface. This preparation has been given the name active nematic (AN)[12]. Under continuous supply of adenosine triphosphate (ATP), a layer of ordered filamentous protein bundles self assembles featuring a turbulent-like behavior. The currently accepted paradigm for active turbulence[3–7] in ANs assumes that vortical flows originate from local stress sources that are associated to orientational gradients within the active material[23–26]. Recent numerical studies[7] suggest that, in spite of the apparently chaotic flow behavior, this regime features an intrinsic length scale, $l_\alpha \sim \sqrt{K/\alpha}$. This scale arises from a balance between active and passive stresses[23–25], and may be experimentally varied by tuning the activity ($\alpha$) or the rigidity ($K$) of the AN.

Within this framework, here we study the mechanism with which an active nematic constrained to flow in a quasi-two-dimensional geometry adapts to lateral confinement exerted by a structured liquid interface. Our analysis is based on providing a quantitative assessment of the geometry of the active flows, comparing the AN in contact with an isotropic oil with the AN flowing in contact with a reconfigurable interface featuring a well-defined mesostructure. In this work, we will call the former the unconstrained configuration while the latter will be the constrained configuration. We begin by presenting clear experimental evidence for an exponential distribution of vortex sizes in the unconstrained active turbulence regime. Next, we show that this geometry is reversibly altered when the active flows are locally captured by mesoscopic circular domains, which impose their scale-free size distribution on the AN. By modifying the activity and the rigidity of the AN, we demonstrate that the active length scale reemerges in the constrained regime as a cutoff domain size for the effective capture of the active flow. Finally, we demonstrate that the prevalence of $l_\alpha$ is due to its intimate relation with the dynamics of topological defects that underlie the active flows.

## Results

**Geometry of unconstrained active turbulence.** The material we consider here is constituted by fluorescently labeled, micron-sized microtubules (MTs), bundled together by the depleting action of

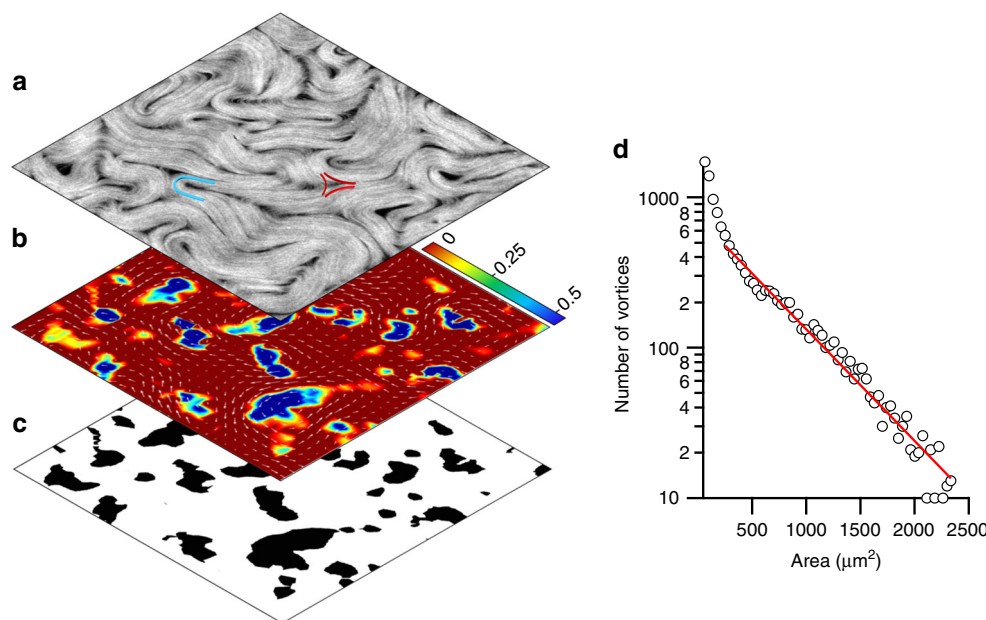

**Fig. 1** Structure of the active turbulent flow in contact with an isotropic oil. **a** Confocal fluorescence micrograph (375 × 375 μm$^2$) of the AN in the turbulent regime when in contact with an isotropic oil. An example of the proliferating +1/2 (*blue*) and −1/2 (*red*) defects is sketched on top of the image (Supplementary Movie 1). **b** Instantaneous flow field (*vector plot*) and computed Okubo–Weiss parameter field (*density plot*, arbitrary units), which is employed to determine the location and size of vortices. **c** Binary image corresponding to the Okubo–Weiss field. **d** Statistical analysis of the distribution of vortex sizes accumulated for a series of one thousand snapshots during the active turbulent flow. The *solid line* is an exponential fit to the data. The range of vortex sizes is limited by the field of view

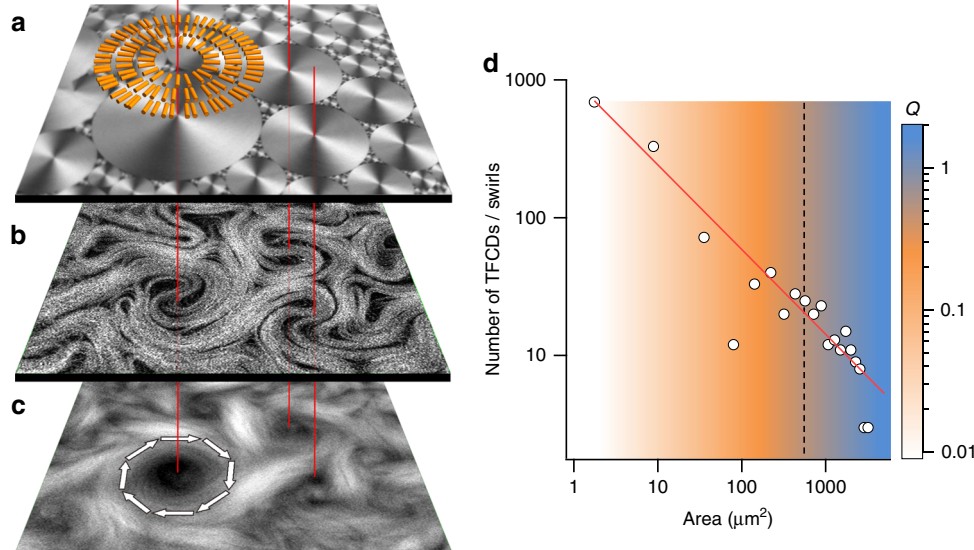

**Fig. 2** Self-assembly of the active nematic in contact with the patterned interface. **a** Confocal reflection micrographs of the water/liquid crystal interface. The diagram illustrates the arrangement of the 8CB molecules at the interface. **b** Confocal fluorescence micrograph of the AN. **c** Time averaged fluorescence confocal micrograph (total integration time 300 s). *Arrows* indicate the direction of the circular flow. *Line segments* through the *panels* identify coincident regions, showing that swirls are in registry with underlying TFCDs. The field of view is 240 × 240 μm² (Supplementary Movie 2). **d** Size distribution of TFCDs in a 1 mm² window. The *red solid line* is a power law fit to the data. The *background colour map* corresponds to experimental values of the winding number $Q$ as a function of domain area. The *dashed vertical line* marks the characteristic area in the size distribution of the unconstrained turbulent regime (Fig. 1)

polyethylene glycol (PEG), and cross-linked by clusters of kinesin motor proteins[12]. Under continuous supply of ATP, the tens-of-microns-long filamentous bundles are subjected to permanent internal stresses due to the action of the motor complexes (see Methods). This aqueous preparation spontaneously self-assembles into a flow-permeated active gel[27–29] that condenses as an AN layer at surfactant-decorated oily interfaces. As a reference experiment, a layer of AN is interfaced with an isotropic oil in a custom-built cell (see Methods). The AN textures that form upon continuous stretching and folding of MT bundles are punctured by disclinations that proliferate under high activity conditions, giving rise to a two-dimensional active turbulent steady state[3–8] (Fig. 1a; also see Supplementary Movie 1). Defects can be easily identified in the micrographs as moving regions devoid of MTs around which active filaments are organized either with a parabolic (+1/2 defects) or a hyperbolic (−1/2 defects) arrangement (Fig. 1a). Because of their symmetry, moving +1/2 defects actively organize the flow, while −1/2 defects are simply advected. This configuration constitutes the active turbulent regime, whose geometry we have characterized by analyzing the statistical distribution of vortex sizes. To locate and measure the area of each vortex, the local instantaneous velocity of the active flow is evaluated from a sequence of micrographs. Velocimetry data are used to obtain values of the Okubo–Weiss parameter, $OW = (\partial_x v_x)^2 + \partial_y v_x \cdot \partial_x v_y$, which provides with a standard criterion for vortex location by considering the extension of each vortex to be bound by the condition $OW < 0$[7, 30] (Fig. 1b, c). A statistical analysis reveals an exponential distribution of vortex sizes (Fig. 1d), from which we can extract a characteristic vortex area, $A^*$, that determines de decay of population as a function of vortex size. For the experimental conditions of the experiment in Fig. 1 ([ATP] = 700 μM, [PEG] = 0.8% (wt/wt), oil viscosity = 12.5 Pa s) we estimate $A^* = 585 \pm 17$ μm². We can, therefore, extract a characteristic length scale for the unconstrained flow geometry $l_a \sim \sqrt{A^*}$ that is known to increase when the AN rigidity increases or when its activity decreases[7]. We can also extract a characteristic time scale for this system by analyzing the lifetime

of individual vortices. To this purpose, we have analyzed the time oscillations of the OW parameter in regions of size $A^*$ (Supplementary Fig. 1), and obtained a well-defined frequency of ~25 mHz, which yields a typical lifetime of ~40 s for vortices in this regime. The experiments reported here have been performed with the AN flowing in contact with a silicon oil with a dynamic viscosity of the same order as the low-shear viscosity of the anisotropic oil described below to better appraise the effect that imposing lateral order has in the active flow geometry.

**Active flow constrained by a patterned interface.** In order to impose lateral order on the AN flow, we take advantage of the hydrodynamic coupling that has been demonstrated to exist between the active and passive fluids across the water/oil interface[31]. Moreover, we take advantage of the giant viscous anisotropy of Smectic-A (SmA) liquid crystal (LC) oils[32] to establish a patterning in the boundary conditions for the active flow. Note that a patterning imposed by means of a solid surface is unpractical in the present case, since the AN has been observed to only form in contact with a liquid interface[12]. The chosen mesogen, octyl-cyanobiphenyl (8CB), features mesophases at temperatures compatible with protein activity (see Methods). At room temperature, free energy minimization results in the SmA phase spontaneously self-assembling into polydisperse domains, known as toroidal focal conic domains (TFCDs)[32], that organize into a fractal tiling known as Apollonian gasket[32, 33] (Fig. 2a). At the water/LC interface, TFCDs have a circular footprint, and are formed by concentric SmA planes perpendicular to the interface. Thus, 8CB molecules, which are both parallel to the water layer due to the interaction with the used surfactant, and perpendicular to the SmA planes, orient radially in concentric rings (Fig. 2a; Supplementary Fig. 2). Crucially, molecules in the SmA phase can diffuse freely within a given smectic plane but their transport is severely hindered in the direction perpendicular to the planes. Although 8CB is, from a macroscopic point of view, a liquid, the local interfacial shear stress probed by the active material is

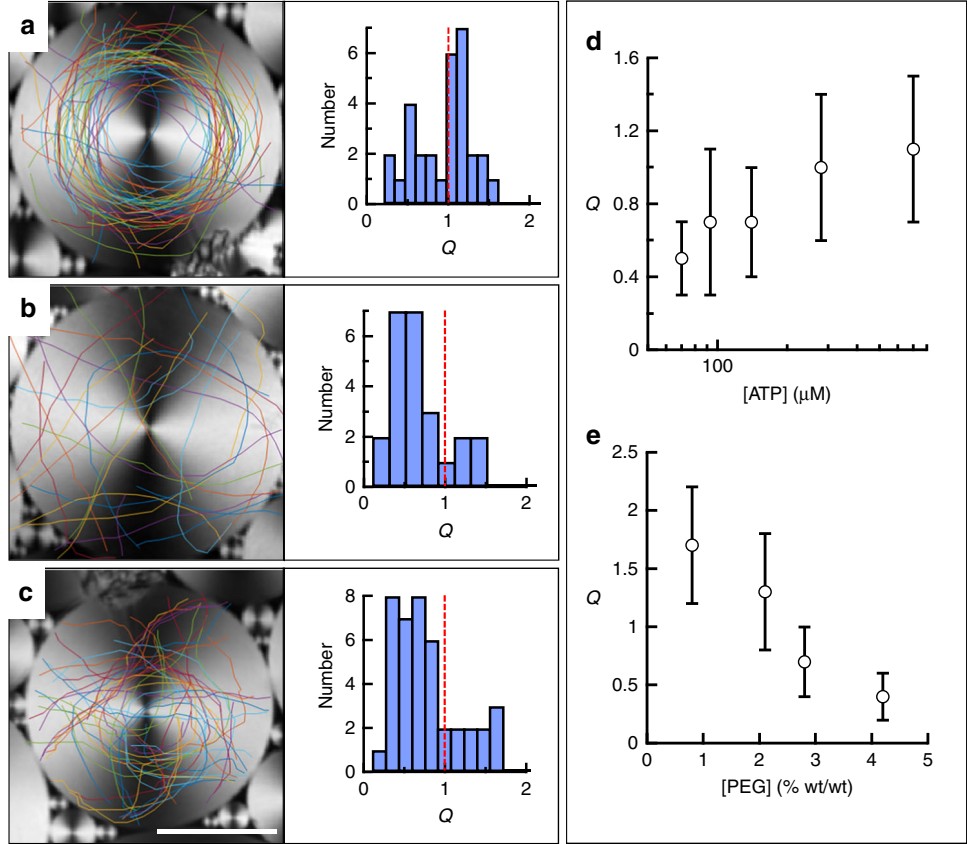

**Fig. 3** Soft confinement of the active nematic. Confocal reflection micrographs of different TFCDs and overlaid trajectories of +1/2 defects of the flowing AN for different experimental conditions: **a** [ATP] = 700 µM, [PEG] = 0.8% (wt/wt); **b** [ATP] = 70 µM, [PEG] = 0.8% (wt/wt); **c** [ATP] = 700 µM, [PEG] = 4.2% (wt/wt). Histograms next to micrographs include the distribution of measured winding numbers, Q. Scale bar 25 µm. Each *line color* corresponds to the trajectory of a different defect (Supplementary Movie 3). **d** Average value and standard deviation of Q for domains with diameter 60 µm, [PEG] = 1.6% (wt/wt), and different values of [ATP]. **e** Average value and standard deviation of Q for domains with diameter 60 µm, [ATP] = 1.4 mM, and different values of [PEG]. *Error bars* are the standard deviation for the measurements under each condition

markedly anisotropic, forcing active stretching of the MT bundles to occur preferentially along circular trajectories, centered in TFCDs. Swirling laminar currents now evolve within the interfacial domain limits, segregated from the rest of the large-scale flows in the system (Fig. 2b, c; also see Supplementary Movie 2). This results in the size distribution of swirls to become commensurate with the Apollonian gasket. The original exponential vortex size distribution in the unconstrained turbulent AN is now transformed into a scale-free power law, matching the distribution of underlying domains, for which we find a better fit with an exponent −0.7 (Fig. 2d).

Our experiments reveal the emergence of a cutoff size above which TFCDs can locally imprint a swirl on the AN, while small domains can, at most, scatter moving defects. To provide with a quantitative assessment of the flow confinement due to a given TFCD, we define, for each +1/2 defect that hovers within the domain's limits, a winding number, $Q = \Delta\alpha/2\pi$, where $\Delta\alpha$ is the accumulated angle of rotation around the TFCD center until the defect is either annihilated or it escapes from the domain. For the reference experimental conditions [ATP] = 700 µM, [PEG] = 0.8% (wt/wt), we have plotted the average value of Q in Fig. 2d (Supplementary Fig. 4a). Our data reveal that domains smaller than 25 µm in diameter do not significantly alter the defect trajectories, exerting an average confinement $Q < 0.5$, while domains larger than 40 µm in diameter exert a confinement $Q > 1$, effectively trapping each passing defect. When comparing the unconstrained and the constrained active flow, we observe

that the characteristic vortex size measured from the exponential distribution in the former (indicated by a line in Fig. 2d) matches with the cutoff domain size observed in the latter.

The effectiveness of the active flow confinement depends both on the size of each TFCD and on the state of the AN. We have quantified the influence on the average winding number of the concentration of ATP, which will directly modify the activity parameter $\alpha \sim \log[\text{ATP}]$, and the bending rigidity (K) of the active bundles, which is expected to increase, albeit with an unknown dependency, with the concentration of the depleting agent PEG[34] (Fig. 3; Supplementary Figs. 3 and 4; Supplementary Movie 3). For a TFCD with diameter 60 µm (Fig. 3a), for which $Q > 1$ under the reference experimental conditions, passing defects are forced to circulate around the domain center, following the circular easy flow directions defined by the underlying SmA phase (Fig. 3a). Accumulated defect trajectories reveal an inner circular region of diameter ~20 µm forbidden for circulating defects. A decrease in $\alpha$ (Fig. 3b, d) or an increase in K (Fig. 3c, e) lead to a shift to smaller Q values for a given domain size, i.e., to an increase in the critical size for domains to capture the active flow.

**Intrinsic length and time scales in the constrained flow.** To understand the origin of the threshold TCFD size for the effective capture of moving filaments, we focus on how the AN self-organizes into swirls. As a characteristic feature of

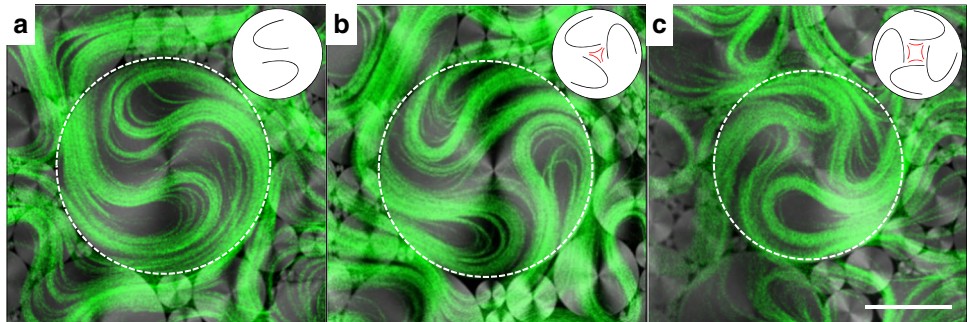

**Fig. 4** Defect structure in active nematic swirls. Confocal micrographs of the AN in contact with the lattice of TFCDs. The simultaneously acquired confocal fluorescence (*green*) and confocal reflection (*grayscale*) channels are overlaid, and show, respectively, the AN filaments and the SmA interface. The contour of the central TFCD is marked with a *white dashed* circumference. Trapped topological defects are illustrated by the *sketches,* and result in a topological charge **a** $S = 2 \times (+1/2)$; **b** $S = 3 \times (+1/2)-1/2$; **c** $S = 4 \times (+1/2)-1$. *Scale bar*, 50 μm. (Supplementary Movie 4)

active turbulence, defects are permanently renovated through instabilities of the elongated filaments[23–25], even their number may change to a small extend, continuously assembling and disassembling the core structure of each rotating swirl. From a topological perspective, the confined rotating filaments constitute a singularity of total charge $S = +1$, which corresponds to a full rotation around the domain center. However, the spontaneous folding of the extensile AN filaments can only create +1/2 defects (at the tip of the fold) and −1/2 defects (at the tail of the fold). This topological mismatch poses an additional constraint on the ensemble of AN defects that may evolve within a TFCD. At all times, a dynamic balance must be established such that the arithmetic sum of defect charges inside a single domain adds up to one, as dictated by topology (Fig. 4; also see Supplementary Movie 4). Clearly, the minimum number of semi-integer defects required to organize a rotating swirl is two $S = +1/2$ defects (Fig. 4a). Since defect spacing is regulated by $l_\alpha$[7, 8, 25, 35, 36], the above topological arguments establish a minimum size for a rotating swirl. Considering that the threshold domain size for effective flow trapping scales with material parameters consistently with $l_\alpha$, we conclude that it is indeed this active length scale that also determines the crossover from turbulent to laminar flow under constrained conditions.

The ability of constraining the active flow allows to easily probe intrinsic time scales in the system. It is known that the extensile nature of the studied AN material makes a configuration of parallel elongated filaments intrinsically unstable, leading to a defect-forming bending instability[23, 24]. As observed in Figs. 2c and 3a, the distribution of the trapped active filaments around large TFCDs is heterogeneous, with a central region mostly devoid of MTs surrounded by a denser region occupied by the outer arms of +1/2 defects. Since filaments are locally parallel within this region, the inner structure of organized $S = +1$ swirls is unstable, being periodically disrupted and reconstituted. An example of this process is depicted in Fig. 5 (Supplementary Movie 5), where we show a swirl, which has been assembled by a TFCD of approximate diameter 200 μm. The vortices in the original turbulent regime have been transformed into swirls of circulating laminar flow. In the first micrograph (Fig. 5a), the AN is structured as an annular band of circularly-aligned MT bundles. This configuration is unstable (Fig. 5b), and the filaments develop a radial buckling instability that leads to the formation of semi-integer defect pairs, moving inwards and outwards of the domain (Fig. 5c, d). Some of these new defects will eventually annihilate with existing inner ones (Fig. 5e, f), finally reorganizing the original circularly-aligned state (Fig. 5g). It is remarkable that the handedness of flow rotation inside each swirl, which is randomly selected upon self-assembly, is preserved

through these episodes of structural instability. This reproducible sequence of dynamic events is also exhibited as a time periodic modulation of the velocity of the active flows constrained inside domains. In Fig. 6a, we show the trajectory of a micro-particle trapped at the SmA/water interface, moving along the circular easy-flow directions of the TFCD, and propelled by coupling with the AN. An analysis of the time evolution of the particle's trajectory, which acts as an active flow tracer, displays a remarkable periodicity (Fig. 6b, c; also see Supplementary Movie 6), thus revealing an intrinsic time scale of the AN[22, 25]. The average period of the oscillations, $T \sim 25$ s, is of the same order as the lifetime of vortices in the unconstrained AN flow.

## Discussion

It is well accepted in the literature that the spatial arrangement of ANs is determined by an intrinsic length scale $l_\alpha$, which establishes spatial features in the material such as the steady-state defect separation and the bending radius of the extensile MT bundles. Earlier simulations revealed that this length scale also determines the geometry of the active turbulent regime[7], where the vortex size distribution follows an exponential whose decay length is related to $l_\alpha$. Our experiments provide quantitative evidence to confirm these predictions when the AN evolves in contact with an isotropic oil (Fig. 1). On the other hand, when flowing in contact with the highly anisotropic SmA phase of 8CB, the active turbulence is reorganized in a scale-free power law distribution of swirl sizes that adapt to the underlying SmA tiling. Even in this regime, an intrinsic length scale resurges in the form of a minimum swirl size inside of which AN flow is laminar. We have provided topological arguments that justify the existence of this minimum size, since at least two +1/2 defects (parabolic folds) of the active filaments are required to organize a circulating swirl, and the defect separation is regulated by $l_\alpha$[7, 8, 25, 35, 36]. Moreover, we have found that the scaling of this characteristic size with the material parameters $\alpha$ and $K$ is consistent with the one expected for $l_\alpha$. This evidence, along with the topological arguments presented above, let us summarize our findings as a manifestation of the different roles played by the same active length scale $l_\alpha$, depending on the confinement constraints.

In addition, the ability to entrap the flows of active materials provides with a useful tool to constrain their dynamics and study new organizational features arising in bounded conditions. In this work, by confining the AN in circulating swirls, we have put into evidence the metastable nature of the extensile active material and revealed the existence of periodic rearrangements in the aligned AN. Active stress builds up, and it is released in periodic bursts of defect creation, followed by annihilation and the recovery of

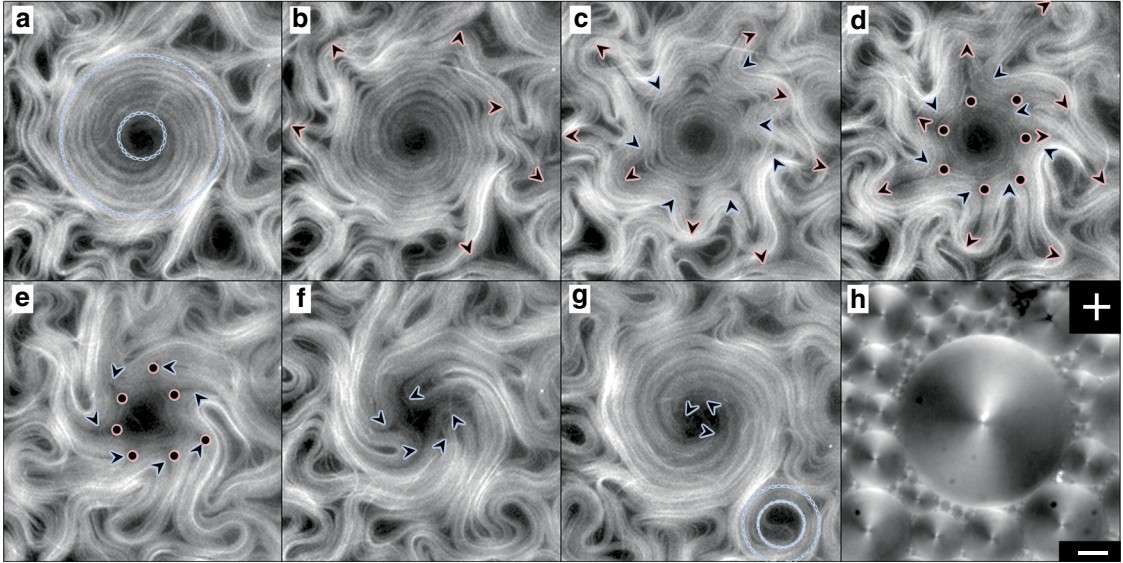

**Fig. 5** Defect dynamics in active nematic swirls. Fluorescence confocal micrographs of the AN swirls constrained by a TFCD (Supplementary Movie 5). Experimental conditions [ATP] = 700 µM, [PEG] = 0.8% (wt/wt). **a** Rotation of +1/2 defects results in the formation of a corona of MT bundles. *Dashed lines* depict inner and outer corona perimeters. **b–g** Evolution of the AN swirl in **a** after it becomes unstable. Elapsed times after **a** are 15 s (**b**), 32 s (**c**) 45 s (**d**), 71 s (**e**), 95 s (**f**), and 136 s (**g**). In **g**, a smaller TFCD has assembled a MT corona. For clarity, −1/2 defects are only highlighted in **d** and **e**. **h** Polarizing micrograph taken between crossed polarizers (*top right corner*) of the local distribution of TFCDs that leads to the active flow patterns in **a–g**. *Scale bar*, 50 µm

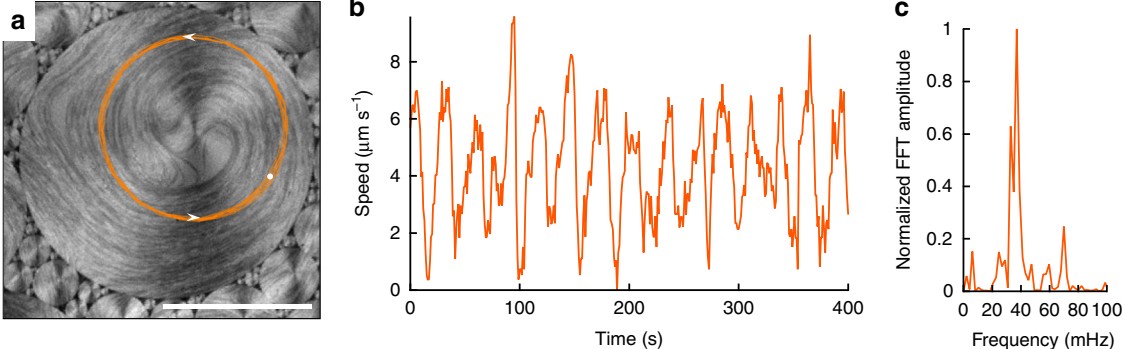

**Fig. 6** Oscillations of the swirl speed. **a** Confocal reflection micrograph with the trajectory of a particle adsorbed at the water/LC interface that is being advected by the active flow (Supplementary Movie 6). **b** Instantaneous speed of the tracer in (**a**). **c** Power spectrum of the data in (**b**). *Scale bar* 50 µm

the aligned state. Our experiments reveal the existence of a well-defined time scale for these oscillations, which is compatible with the lifetime of vortices in the unconstrained active turbulence regime. One could define a time scale, $\tau$, by combining the average +1/2 defect speed, $v$, and the intrinsic active length scale, so that $\tau \sim l_\propto / v$. Although we have shown that the same intrinsic length scale emerges regardless on the confinement constraints, it is unclear whether the defect speed should be affected by confinement and, therefore, what should be the expected scaling behavior of $\tau$.

The ability to harness active turbulence has allowed in this work to organize well-defined rotating flows, which might have the potential to act as microscale machines. As a proof-of-concept, we have seeded the active material with passive polystyrene micro-beads, which are dragged by the active flow (Supplementary Movie 6). As they are not bound to a circular trajectory, tracers inside rotating swirls drift outwards while performing spiral trajectories. However, particles can be adsorbed

at the LC phase, thus describing perfect circular trajectories that follow the geometry of the SmA planes.

In summary, our experiments have demonstrated that constrainment with an anisotropic soft interface can lead to the modification of the nature and geometry of active flows, whose structure is nevertheless always determined by the same single intrinsic length scale. It is the role of this length scale, and not its value, that is affected by the confinement conditions. This observation overturns our idea of active flows as it confers them with a minimal regularity, and permits to envisage strategies of control vastly different from those applying to their passive counterparts. In addition, the adaptability to soft confinement provides with a handle to probe the intrinsic length and time scales of active materials, and paves the way for further studies comparing classical and active two-dimensional turbulence. Moreover, our findings may unveil an alternative mechanism by which cells self-regulate and control cytoplasmic flows through reconfigurable membranes. In fact, the role of topology

## Methods

**Lamellar LC**. 4-cyano-4′-octylbiphenil (8CB, Synthon; ST01422) is a thermotropic liquid crystal between 21.4 and 40.4 °C, featuring a lamellar Smectic-A phase in the temperature range 21.4 °C < T < 33.4 °C.

**Polymerization of MTs**. Heterodimeric (α,β)-tubulin (from bovine brain, obtained from the Brandeis University Biological Materials Facility) is incubated at 37 °C for 30 min in an M2B buffer (80 mM PIPES, 1 mM EGTA, 2 mM MgCl$_2$; Sigma; P1851, E3889 and M4880, respectively) supplemented with the reducing agent dithiothrethiol (Sigma; 43815) and with Guanosine-5′-[(α,β)-methyleno]triphosphate (GMPCPP; Jena Biosciences; NU-405), a non-hydrolysable analogue of the nucleotide GTP. As GTP, GMPCPP promotes the association between tubulin heterodimers although it completely suppresses the dynamic instability of MTs[39]. By controlling the concentration of GMPCPP we prepare high-density suspensions of short MTs (1–2 µm). For the characterization with fluorescence microscopy, part of the initial tubulin (3%) is fluorescently labeled.

**Kinesin expression**. *Drosophila melanogaster* heavy chain kinesin-1 K401-BCCP-6His (truncated at residue 401, fused to biotin carboxyl carrier protein (BCCP) and labeled with six histidine tags) has been expressed in *Escherichia coli* by using the plasmid WC2 from The Gelles Laboratory (Brandeis University) and purified with a nickel column. After dialysis against 500 mM Imidazole buffer, kinesin concentration is estimated by means of absorption spectroscopy and stored at a specific concentration in a 40% (wt/vol) sucrose solution at −80 °C[40].

**Preparation of molecular motor clusters**. Biotinylated kinesin motor proteins and tetrameric streptavidin (Invitrogen; 43-4301) are incubated on ice for 30 min at specific stoichiometric ratio (~2:1) to obtain kinesin-streptavidin motor clusters.

**Assembly of the active gel**. MTs are mixed with the motor clusters that act as cross-linkers, and with ATP (Sigma; A2383) that drives the activity of the gel. The aqueous dispersion contains a non-adsorbing polymer (PEG, 20 kDa) (Sigma; 95172) that promotes the formation of filament bundles through depletion interaction (Supplementary Fig. 2). To maintain a constant concentration of ATP during the experiments, an enzymatic ATP-regenerator system is used, consisting on phosphoenolpyruvate (Sigma; P7127) that fuels pyruvate kinase/lactate dehydrogenase (PK/LDH) (Invitrogen; 434301) to convert ADP back into ATP. Several antioxidant components are also included in the solution to avoid protein denaturation, and to minimize photobleaching during characterization by means of fluorescence microscopy. The PEG-based triblock copolymer surfactant Pluronic F-127 (Sigma; P-2443) is added at 1% (wt/wt) (final concentration) to procure a biocompatible water/oil interface in subsequent steps.

**Experimental Setup**. The studied AN is formed at the interface with either an isotropic silicone oil (BlueStar Silicones; BlueSil® V47:12500) or 8CB. The interface is prepared in a cylindrical well of diameter 5 mm and depth 2 mm, manufactured with a block of poly-dimethylsiloxane (Sylgard®, Dow Corning) using a custom mold. The block is glued onto a bioinert and superhydrophilic polyacrylamide (PAA)-coated glass, which is prepared following reported protocols[41] (Supplementary Fig. 5). In brief, clean and activated glass is first silanized with an acidified ethanolic solution of 3-(Trimethoxysilyl)propylmethacrylate (Sigma; 440159), which will act as polymerization seed. The silanized substrates are subsequently immersed in a aqueous solution of acrylamide monomers (Sigma; 01697) (2% (wt/vol), for at least 2 h) in the presence of the initiator ammonium persulphate (Sigma; A3678) and N,N,N′,N′-Tetramethylethylenediamine (Sigma; T7024), which catalyzes both initiation and polymerization of acrylamide. The cavity is first filled with ~30 µl of oil and, subsequently, 1 µl of the water-based active gel is injected between the hydrophobic oil and the superhydrophilic glass plate (Supplementary Fig. 5). Pluronic F-127 stabilizes the interface and avoids direct contact between the oil and the protein-based active mixture. Moreover, this surfactant ensures a planar alignment of 8CB molecules at the water/LC interface. When 8CB is used, samples are heated up to 35 °C to promote transition to the less viscous nematic phase of the LC, which facilitates the spreading of the active gel onto the PAA-coated substrate. Temperature is controlled by using a thermostatic oven built with Thorlabs SM1 tube components and foil heater, which is regulated with a Thorlabs TC200 controller. After several minutes at room temperature, the active material in the gel spontaneously condenses onto the flat water/oil interface, leading to the formation of the AN layer. Unlike conventional flow cells, in which a layer of the active gel is confined in a thin gap between two glass plates[12], our open setup enables us to prepare the interface using oils with a high viscosity.

**Imaging**. Routine observations of the AN were performed by means of conventional epifluorescence microscopy. We used a custom-built inverted microscope with a white led light source (Thorlabs MWWHLP1) and a Cy5 filter set (Edmund Optics). Image acquisition was performed with a QImaging ExiBlue cooled CCD camera operated with ImageJ µ-Manager open-source software. For sharper imaging of the interfacial region, we used a Leica TCS SP2 laser-scanning confocal microscope equipped with a photomultiplier as detector and a HeNe–633-nm laser as light source. A ×20 oil immersion objective was employed. We performed confocal acquisition in fluorescence and reflection modes, simultaneously.

**Image analysis**. Tracer-free velocimetry analysis of the AN was performed with a public domain particle image velocimetry (PIV) program implemented as an ImageJ plugin[42]. Flows were also traced by dispersing PEGylated spherical polystyrene microbeads of diameter 1.7 µm (Micromod; 08-56-173). Manual Tracking ImageJ plugin was used to manually track trajectories of particles or defects in the AN. Further analysis of velocimetry data was performed with custom-written MatLab codes. In order to quantify the vortex distribution in the turbulent AN, the Okubo–Weiss (OW) parameter was mapped from the velocity fields obtained by image velocimetry (Fig. 1b). The vortex areas were quantified from thresholded OW field binary images in ImageJ (Fig. 1c). Vortices on the edges of the field of view were excluded.

**Data availability**. All data used in the preparation of this manuscript and the Supplementary Material are available on request from the authors.

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

## Acknowledgements

The authors are indebted to Z. Dogic and S. DeCamp (Brandeis University), and the Brandeis University MRSEC Biosynthesis facility for their assistance in the preparation of the active gel. We thank B. Hishamunda (Brandeis University), and M. Pons, A. LeRoux, and G. Iruela (Universitat de Barcelona) for their assistance in the expression of motor proteins. We thank J. Hardoüin for his assistance in some of the experiments. We thank R. Casas and G. Valiente (Bluestar Silicones) for providing the silicone oil sample. We thank J. Casademunt (Universitat de Barcelona) and O.D. Lavrentovich (Kent State University) for fruitful discussions. Funding has been provided by MINECO (Projects FIS 2013-41144P and FIS2016-78507-C2-1-P AEI/FEDER-EU). P.G. acknowledges funding from Generalitat de Catalunya through a FI-DGR PhD Fellowship. Brandeis University MRSEC Biosynthesis facility is supported by NSF MRSEC DMR-1420382.

## Author contributions

All authors conceived the research. P.G. performed the experiments and data analysis. J.I.-M. and F.S. supervised the research and wrote the manuscript. All authors discussed the results.

## Additional information

**Competing interests:** The authors declare no competing financial interests.

