## [Peer Review File · Nature Communications]

REFeree REPORT

Manuscript#: NCOMMS-17-05073

Corresponding Author: Jordi Ignés-Mullol

Title: Taming active turbulence with patterned soft interfaces

The authors report an experimental study of self-sustained turbulence in a microtubule-based active nematic fluid constrained in a quasi-2D geometrical setup. The active fluid resides on a bio-inert fluid substrate consisting of a passive fluid phase. This may either be an oil phase with a simple, isotropic structure or a smectic liquid crystal with a planar conical mesostructure. The latter imparts an intricate static circular pattern (resembling an Apollonian fractal) onto the turbulent flow of the microtubule bundles, thereby forcing the vortical flow to comply with the circular geometry of the large conical domains but only if the swirl size exceeds a certain cut-off dictated by the $S=+1$ topological charge of the swirl. A subsequent analysis of the defect mobility at different experimental conditions (tubule depletion strength, motor protein activity) suggests that the cut-off length for the pinned swirls has the same origin as the length scale that sets the typical vortex size of the unbounded active fluid. This length scale is intrinsic and is set by the ratio of active stresses generated by, in this case, the kinesin motor proteins and the (passive) Frank elastic stresses experienced by the nematic microtubule bundles. The study also highlights the dynamics of the internal defect break-up and reassembly of the pinned nematic swirls. The defect dynamics exhibits a remarkable degree of periodicity suggesting a typical time scale that needs to be further explored.

I find the results reported in this manuscript quite exciting and I would certainly recommend this work for publication in Nature Communications. There are three reasons for this. First, the idea of 'taming' active turbulent flow using patterned liquid crystalline fluid substrates is novel and might lead to a number of useful microfluidic applications in the future. Second, the main physical message of this work -- there is an intrinsic length scale (ratio of active versus nematic stresses) for unperturbed active flow which re-expresses itself as a cut-off length in patterned flow conditions -- is an unexpected and important finding which sheds new light on how active fluids respond to external stimuli. Third, the setup provides valuable data on vortex size distributions, defect dynamics, etcetera, obtained under carefully controllable experimental conditions. These data may, for instance, be used to test and (in)validate various theoretical active fluid models. This is an important asset in the field of active matter physics where theoretical and numerical studies have largely outpaced their experimental counterparts.

The paper is written in a clear and succinct style. As far as I can judge the analysis is carefully done and the results are (mostly) clear.

Below I formulate a number of queries I'd request the authors to clarify prior to further consideration for publication:

1) How static are the TFCF patterns? Do the 'rings' show any appreciable rearrangement on the typical timescale of a measurement? Does the active flow on top of the smectic fluid affect the rigidity of the Apollonian pattern in any way?

2) The length scale associated with the exponential distribution in Fig. 1d is not given explicitly. How does this value compare to the cut-off value implied by the shaded region in Fig. 2d? If possible, this should be indicated more explicitly in the viewgraphs.

3) The manuscript would also greatly benefit from a brief discussion of the possibility (or impossibility) to estimate the typical magnitude of the activity coefficient (α , which I presume has units energy per volume) and rigidity parameter K (units force) from molecular principles, perhaps by referring to previously published experimental data on these systems.

4) Figure 3 is somewhat difficult to interpret as the defect trajectories shown correspond to different experimental conditions (ATP/PEG concentrations). I think the evidence for the sensitivity of the intrinsic length with respect to a change of kinesin activity and bundle rigidity (via polymer depletion) needs to be laid out more systematically, for instance by exploring the maximum of Q versus ATP or PEG concentrations.

5) The authors mention (in Figure S3) that the pinned nematic swirls tend to 'assemble and disassemble with great regularity' suggesting a typical time scale/frequency. This naturally begs the questions; what sets this time scale and how does it relate to the typical vortex lifetime in an unbounded active flow? A deeper understanding of the origin of these time scales would certainly enhance the scope of the manuscript.

Reviewers' comments:

Reviewer #2 (Remarks to the Author):

The manuscript presents studies on active matter on top of a smectic liquid crystals. Both the topic and results are interesting. However, the manuscript is not well written and needs significant improvements. The following are some comments:

(1) The authors often mixed main texts and figure captions (in both the main manuscript and supporting material), making not only hard to follow the results in the main text but also the figure captions very complicated.

(2) The abstract does not describe the main research and findings well. Especially, the last sentence is difficult to understand.

(3) Between line 41 and 61, the authors mixed introduction and their research. After reading it a couple of times, I still did not get what is the motivation of this research, and what is the new findings in this research, and what is its importance or novelty.

(4) In subsections from 64-156, each sections have only one paragraph. The authors again mixed introduction to references with descriptions of their results. It seems there is no simple logic follow.

(5) In Fig 2d, the data points are quite scattered. Since the conclusion of scale free flow is based purely on this figure, the authors should show that the relationship is nothing other than scaling law. For example, the authors at least need to show that the original exponential relation observed on smooth oil substrates does not apply here.

(6) Technically, the effects of the smectic liquid crystal substrates need to be explained more clearly. Considering the active layer is only $\sim 100\text{nm}$, the roughness due to the TFCD defects may be important to the active flows. I suggest that authors take some atomic force microscopic images of the liquid crystal substrates, which will be helpful in understanding the effect of the substrate. In addition, the authors used a number of different words (bound, geometric confinements and soft confinements) to describe the substrate effects, which seems to be in conflict with authors' idea that the effects of liquid crystal substrate originates from anisotropic follow resistance. These words are a little too generic, not specific to the system used here.

(7) The authors did not specify how they know the active layer is $\sim 100\text{nm}$, and should add where the estimation comes from in methods section.

(8) For the section of periodic bending instability, there is not specific time for the pictures shown in Fig. 5, how can one see it is periodic? It might be better to have some graphs like Fig. S3c in Fig. 5.

Based on these issues, I recommend that a major revision is necessary.

We thank the reviewers for their thorough revision and useful comments that have helped to improve the manuscript. Below, we provide with an itemized reply to all the comments and suggestions. For clarity, we include the original remarks by the reviewers in blue Times typeface, and our response in black Arial typeface. We clearly indicate, at the end of each response, the corresponding changes in the revised manuscript.

In the manuscript, text that has been modified or added is typed in blue typeface.

Reviewer # 1

We thank the reviewer for the favorable review and for the excellent summary of our work, highlighting in a very clear manner the main arguments to justify the general interest of our work. This has been of great help to clarify the message from our work in the discussion section of the revised manuscript.

Below I formulate a number of queries I'd request the authors to clarify prior to further consideration for publication:

1) How static are the TFCD patterns ? Do the 'rings' show any appreciable rearrangement on the typical timescale of a measurement ? Does the active flow on top of the smectic fluid affect the rigidity of the Apollonian pattern in any way ?

TFCDs are very rigid and static. Only minor rearrangements are observed during the flow of the AN, for observation periods that may elapse tens of minutes. Even in those cases, we believe such rearrangements are most likely due to macroscopic flow in the cell (for instance, due to local dewetting events). In summary, all observations suggest that we can regard the Apollonian pattern as a static pattern, with no appreciable feedback from the flowing AN.

2) The length scale associated with the exponential distribution in Fig. 1d is not given explicitly. How does this value compare to the cut-off value implied by the shaded region in Fig. 2d ? If possible, this should be indicated more explicitly in the viewgraphs.

We thank the reviewer for this suggestion.

Action taken: We have included further details for the reference experiment, namely, unconstrained active turbulent flow of the active nematic in contact with an isotropic oil. We have provided the value of viscosity of the isotropic oil, which was chosen to match the average shear viscosity of the SmA phase, and we have included the length scale (or, similarly, the vortex area scale) obtained from the data analysis. Moreover, this size is explicitly indicated as a vertical line in Fig. 2d (and discussed in the text) to highlight the good quantitative agreement with the critical domain size under the constrained flow conditions.

3) The manuscript would also greatly benefit from a brief discussion of the possibility (or impossibility) to estimate the typical magnitude of the activity coefficient (α , which I presume has units energy per volume) and rigidity parameter K (units force) from molecular principles, perhaps by referring to previously published experimental data on these systems.

This is a most interesting issue whose discussion demands some considerations.

1. The parameter α , correctly identified by the referee as having units of energy per volume, is considered as an effective parameter that accounts for the ATP-based activity of the sample (see ref. 7: L. Giomi et al., Phys. Rev. X 5.031003 (2015)). However, the precise dependence of α on ATP concentration is still under debate. For instance, the original results of T. Sanchez et al., Nature 491, 431 (2012) (ref. 12) are analyzed by S. P. Thampi et al., PRL 111, 118101 (2013) (ref. 4), suggesting that the activity parameter should scale logarithmically with ATP concentration, which agrees with our own observations. Nevertheless, no definitive conclusion can be reached so far.

2. On what respects to the connection of α with first principles-based models, the situation is not clear either. The classic theory of active gels introduces an active contribution to the total stress, proportional to the (tensorial) nematic order parameter, whose coefficient is assumed proportional to the chemical potential difference between ATP and its reaction products (see refs. 17-19: K. Kruse et al., EPJE 16, 5 (2005); F. Julicher et al., Physics Reports 449, 3 (2007); J. Prost et al., Nature Physics 11, 111 (2015)). More hydrodynamically oriented models such as those proposed to describe a two-component active suspension go a little bit further by incorporating some dependence (linear or more often quadratic) of the active stress on the concentration of active units (L. Giomi et al., PRL 106, 218101 (2011); E. Tjhung et al. PNAS 109, 12381 (2012)). Still a way to directly link the “chemical ATP” contents to the activity levels of the active fluid is not clear.
3. Coarse graining microscopic models of active fluids, used to extract useful effective parameters, have also been published (A. Ahmadi et al., PRE 74, 061913 (2006); T. B. Liverpool et al., in Cell Motility, ed. P. Lenz, Springer, NY, (2007)). More recently, these efforts have been continued by using Brownian dynamics-kinetic Monte Carlo schemes in multiscale theories of microtubules and motor-protein assemblies (T. Gao et al. PRL 114, 048101 (2015); see also M.J. Shelley, Annu. Rev. Fluid. Mech. 48, 487 (2016)). In this way one may find expressions for active forces/stresses based on motor parameters, such as velocities or typical displacements. However, again there is no simple way to track the final dependence on the ATP concentration.
4. One must remember that the real active system is based on the collective effects of many motors that crosslink and internally shear microtubules. In this sense, information on individual motor characteristics is certainly valuable, but far from being a definitive clue to the activity level of a condensed sample of active material in terms of the ATP contents.
5. A similar argument also applies when considering material properties (stiffness for instance) of bundled microtubules, the basic textural unit of our active nematics, expected to be different from those of single microtubules. As a matter of fact, very elegant experiments were published recently by the same group that introduced the active nematic material (Z. Dogic group), with the aim of resolving mechanical characteristics (sliding properties, cohesion features) of bundled microtubules (A. Ward et al. Nature Materials, 14, 583 (2015); F. Hilitsky et al. PRL 114, 138102 (2015)).

Actions taken: the effect of activity (through ATP concentration) and rigidity (through PEG concentration) in the constrained flow regime has been discussed in more detail in the revised manuscript, including new data. Unfortunately, there is no evident way to relate α and K to material characteristics of individual motor and filamentous proteins. Further studies will be needed to address these issues.

4) Figure 3 is somewhat difficult to interpret as the defect trajectories shown correspond to different experimental conditions (ATP/PEG concentrations). I think the evidence for the sensitivity of the intrinsic length with respect to a change of kinesin activity and bundle rigidity (via polymer depletion) needs to be laid out more systematically, for instance by exploring the maximum of Q versus ATP or PEG concentrations.

Actions taken: we have performed additional series of experiments where the trapping efficiency of SmA domains, quantified in terms of the winding number Q , is systematically studied as a function of domain size, [ATP], and [PEG]. The main text has been modified to discuss the new data. The data of Q vs domain size has been included as a color map in the updated Figure 2. The dependency of Q on [ATP] and [PEG] has been included in the updated Figure 3. The wide-field-of-view micrographs originally in Figure 3 have been moved to the new Figure S3.

5) The authors mention (in Figure S3) that the pinned nematic swirls tend to 'assemble and disassemble with great regularity' suggesting a typical time scale/frequency. This naturally begs the questions; what sets this time scale and how does it relate to the typical vortex lifetime in an unbounded active flow? A deeper understanding of the origin of these time scales would certainly enhance the scope of the manuscript.

We thank the reviewer for these suggestions. Although the study of this time scale was not a central issue in the original manuscript, we recognize that our system is particularly amenable to carry out such a study. For this reason, we are presenting new data to analyze the vortex lifetime in the unconstrained regime, and we have promoted to the main text some material in the original “supplementary information” where we quantify the time scale in the constrained geometry.

Actions taken: We have included an analysis of the vortex lifetime as Figure S1 of the new Supplementary Information, and have included the result of this study in the main text. Moreover, we have increased the visibility of the analysis that reveals the intrinsic time scale in the constrained regime by moving the material originally in Fig. S3 into Fig. 5 of the revised manuscript. Moreover, we have also compared both time scales in the revised manuscript, and we have discussed about the origin of these time scales in the discussions section.

Reviewer #2 (Remarks to the Author):

The manuscript presents studies on active matter on top of a smectic liquid crystals. Both the topic and results are interesting. However, the manuscript is not well written and needs significant improvements. The following are some comments:

We thank the reviewer for the detailed comments and suggestions that have greatly assisted us in improving the clarity of our original manuscript, as explained below.

(1) The authors often mixed main texts and figure captions (in both the main manuscript and supporting material), making not only hard to follow the results in the main text but also the figure captions very complicated.

Actions taken: We have thoroughly revised the figure captions, both in the main text and in the supporting material, and we have eliminated all text that is non-essential to describe the displayed data or that discusses the data, so that captions are streamlined in the revised manuscript. All data discussions are moved to the main text. Repetition between figure caption and main text have been eliminated.

(2) The abstract does not describe the main research and findings well. Especially, the last sentence is difficult to understand.

Actions taken: We have revised the abstract to improve its clarity and to better convey the main research findings of our work.

(3) Between line 41 and 61, the authors mixed introduction and their research. After reading it a couple of times, I still did not get what is the motivation of this research, and what is the new findings in this research, and what is its importance or novelty.

Actions taken: We have rewritten most of the introduction section of the original manuscript to better explain the motivation for this research, the state-of-the art, and finish with a paragraph that provides with a detailed account of the main new findings of our work.

(4) In subsections from 64-156, each sections have only one paragraph. The authors again mixed introduction to references with descriptions of their results. It seems there is no simple logic follow.

Actions taken:

We have streamlined the main text to clearly separate our results from reference to previous literature. As described above, we have included an account of the main findings of our work at the end of the introduction section, to better guide the readers. Moreover, we have merged sections to make the line of thought of our manuscript easier to follow. In the revised manuscript, the results section is organized in three subsections only:

- A first section where we provide with an experimental analysis of the geometry of the unconstrained active turbulence, obtained when the active nematic flows in contact with an isotropic oil. We have included a succinct account of the main features of this system, including a reference for the crucial role of defect dynamics, which will be essential to explain the main results of our work later in the manuscript. This active turbulent state is like the one that has been described in the literature, and for which there have been some scaling predictions based on numerical modelling. Here, we show, experimentally, that the geometry of the active turbulence can be characterized by an intrinsic vortex size or, equivalently, by an intrinsic length scale that is known to stem from a ratio between nematic and active stresses. This will be the reference state that we will compare when the active flow evolves in contact with a patterned interface.
- A second section where we describe and characterize the active turbulence constrained by the patterned smectic interface. We describe the changes in the geometry of the active flow, and the

correlation with the distribution of the underlying lattice of circular domains. We highlight the main result, namely, the existence of an intrinsic length scale despite the observation of a scale-free power law size distribution. Finally, we study the dependence of this length scale (the minimum domain size for effective flow constraintment) with the available control parameters: ATP and depleting agent concentration, and find a scaling behavior reminiscent of the one reported for the active length scale of the unconstrained flow.

- A third section where we provide experimental evidence to understand the origin of the intrinsic length scale that characterizes the constrained turbulent regime, and show that it emerges naturally because of the dynamic of the topological defects that are known to determine active nematic turbulence. Moreover, we take advantage of the geometrical confinement of the active flow provided by the underlying SmA domains to clearly reveal an intrinsic time scale, that results from the periodic relaxation of stresses that build up due to the extensile nature of the active filaments.

(5) In Fig 2d, the data points are quite scattered. Since the conclusion of scale free flow is based purely on this figure, the authors should show that the relationship is nothing other than scaling law. For example, the authors at least need to show that the original exponential relation observed on smooth oil substrates does not apply here.

Following the reviewer's suggestions, we have improved the statistical analysis in Fig 2d. Originally, only data from a single field of view was analyzed. This time we have scanned the sample and combined different fields of view, for a total area of about 1mm². We believe that the power law fit is quite robust and a much better fit than the exponential distribution found for the case of AN flow in contact with isotropic oils. In the attached plot, we include the updated data (black empty circles), the power law fit (black line) and two attempts at an exponential fit. Red line: exponential fit including all data points. Blue line: exponential fit including only areas above 100 μm². In all cases, the power law fit is consistently better at providing a trend for our data.

Actions taken: we have analyzed an increased number of domains to have a better statistic on the size distribution. The updated Fig. 2d contains the updated data and analysis.

(6) Technically, the effects of the smectic liquid crystal substrates need to be explained more clearly. Considering the active layer is only ~100nm, the roughness due to the TFCD defects may be important to the active flows. I suggest that authors take some atomic force microscopic images of the liquid crystal substrates, which will be helpful in understanding the effect of the substrate.

In regards to the thickness of the active layer, please see our response to point (7) for a detailed account on how we have accurately determined this thickness to be 2.5±0.3μm.

Concerning the roughness of the TFCD defects, their effect on the active flows, and the measurement of such roughness using AFM, we would like to point out that such effect will be important along a plane parallel to the SmA planes, but not so much on a surface perpendicular to the SmA planes [see, for instance Choi et al, PNAS 101, 17340-17344 (2004) ; V. Designolle *et al.*, *Langmuir* 22, 363-368 (2006)]. In our experiments, the SmA molecules are perpendicular to the SmA/air interface, which is tens, even hundreds of micrometers away from the SmA/water interface, where SmA molecules lay parallel to the surface, and where the AN is found. The SmA planes are thus parallel to the SmA/air interface but perpendicular to the SmA/water interface. In other words, we expect the roughness of the TFCD defects to be important at the SmA/air interface, but not important at the SmA/water interface and, therefore, should exert no significant influence on the active flow.

In addition, the authors used a number of different words (bound, geometric confinements and soft confinements) to describe the substrate effects, which seems to be in conflict with authors' idea that the effects of liquid crystal substrate originates from anisotropic follow resistance. These words are a little too generic, not specific to the system used here.

We thank the reviewer for bringing this inconsistency in the language to our attention.

Actions taken: we have thoroughly revised the manuscript and we have adhered to a coherent notation, where we refer to the flow of the active nematic in contact with an isotropic oil as the **unconstrained** regime, and the flow of the active nematic in contact with the patterned SmA liquid crystal as the **constrained** regime.

(7) The authors did not specify how they know the active layer is $\sim 100\text{nm}$, and should add where the estimation comes from in methods section.

We apologize for the confusion caused by our reference to the active layer thickness as being “hundreds” of nm. In fact, the layer thickness depends on the viscosity of the oil across the interface, being thicker for more viscous oils. For the oil viscosity used in this work ($12.5\text{ Pa}\cdot\text{s}$, both for the isotropic oil used in the turbulent regime, equivalent to the average viscosity of the SmA used in the constrained regime experiments) we have performed precise measurements using fluorescence confocal microscopy with a laser-scanning system and a 60x oil immersion objective with $\text{NA}=1.32$. With this instrument, the uncertainty in the thickness estimation has been calibrated to be $0.6\ \mu\text{m}$.

The image below corresponds to an average intensity of 300 XZ cross sections across the AN interface obtained over a 15 min period. We estimate the film thickness as the FWHM of the average intensity profile. The result is a thickness of $2.5\pm 0.3\ \mu\text{m}$.

Fluorescence confocal microscopy across the AN layer. FOV is $80\times 80\ \mu\text{m}^2$.

Intensity profile obtained from the confocal data. FWHM yields the estimated layer thickness.

Action taken: Since it is not central to the discussion of the present work, and it is a matter still under further investigation, we have removed any numerical reference to the estimated thickness of the AN layer from the revised manuscript.

(8) For the section of periodic bending instability, there is not specific time for the pictures shown in Fig. 5, how can one see it is periodic? It might be better to have some graphs like Fig. S3c in Fig. 5.

We thank the reviewer for this suggestion.

Action taken: We have included a time reference in the caption of Fig. 5 and we have included the material in the original Fig. S3 in the updated Fig. 5, since it provides with a clear evidence for the emergence of a periodic behavior in the constrained AN. Discussion of this moved material is enhanced in the revised manuscript.

REVIEWERS' COMMENTS:

Reviewer #1 (Remarks to the Author):

The authors have extensively revised their manuscript and provided additional detail to clarify some of the issues raised in the report.

I have reviewed the revised manuscript and author comments and am quite satisfied with the updated version (in particular regarding the additional information compounded in Figures 2d and 5) as well the with the clarifying remarks in the response letter. I am happy to recommend publication in its present form.

Upon rereading the title I was left wondering a bit as to why the authors chose the term 'taming' rather than e.g. 'manipulating'. Suggesting a weakening or curbing of vortices of some sort one might argue that the imposed Apollonian pattern in fact enhances turbulent flow given that the scale-free vortex distribution (Fig. 2d) is probably more akin to that of a high-Re passive fluid than an overdamped low-Re active fluid.

This, of course, is a purely semantic issue that I would just leave the authors as a mere suggestion.

Reviewer #2 (Remarks to the Author):

The authors have made significant improvements and responded satisfactorily to my comments, I recommend it for publication. Two minor comments:

(1) Some of the sentences are too long. It should make the article more readable if the authors rephrase some of the sentences.

(2) The grayscale image is not very visible in the overlaid image. The authors should try to redo the overlay for better visibility.

Response to the reviewer's report for "Taming active turbulence with patterned soft interfaces", by Guillamat *et al.*

We thank the reviewers for their revision of the updated manuscript, their useful comments, and their final recommendation that our paper be accepted. Below we provide with an itemized response to the issues raised by the reviewers in their final report.

For clarity, we include the reviewer's comment in blue Arial typeface, and our response in black Times typeface.

Reviewer #1 (Remarks to the Author):

The authors have extensively revised their manuscript and provided additional detail to clarify some of the issues raised in the report.

I have reviewed the revised manuscript and author comments and am quite satisfied with the updated version (in particular regarding the additional information compounded in Figures 2d and 5) as well the with the clarifying remarks in the response letter. I am happy to recommend publication in its present form.

Upon rereading the title I was left wondering a bit as to why the authors chose the term 'taming' rather than e.g. 'manipulating'. Suggesting a weakening or curbing of vortices of some sort one might argue that the imposed Apollonian pattern in fact enhances turbulent flow given that the scale-free vortex distribution (Fig. 2d) is probably more akin to that of a high-Re passive fluid than an overdamped low-Re active fluid.

This, of course, is a purely semantic issue that I would just leave the authors as a mere suggestion.

We thank the reviewer for this suggestion. When using the word "taming" we want to convey the idea that, despite the intrinsically chaotic dynamics of the active material, one may somehow change its structure by contact with a suitable patterned interface. The fact that, in the present work, our interface contains an Apollonian pattern with a power law domain size which is transferred to the original vortex distribution does not necessarily mean that the resulting active flow is closer to the high-Re turbulence. Because of this, we feel that our original choice of title is more suitable.

Reviewer #2 (Remarks to the Author):

The authors have made significant improvements and responded satisfactorily to my comments, I recommend it for publication. Two minor comments:

(1) Some of the sentences are too long. It should make the article more readable if the authors rephrase some of the sentences.

We thank the reviewer for this suggestion. We have revised the manuscript and rephrased long sentences, often splitting them into two separate sentences.

(2) The grayscale image is not very visible in the overlaid image. The authors should try to redo the overlay for better visibility.

We understand the reviewer refers to Fig. 4. We have optimized the contrast in the background grayscale image, which is more clearly observed in the included Supplementary Movie 4. Moreover, we have highlighted with a circumference the contour of the central TFCd where we study the topological charge balance of defects in the trapped active flow.